# Promoting HPV Vaccination in People with HIV: Factors to Consider

**DOI:** 10.3390/ijerph20075345

**Published:** 2023-03-31

**Authors:** Kinsey A. Huff, Ashlea Braun, Michelle R. Salvaggio, Patrick McGough, Summer G. Frank-Pearce, Darla E. Kendzor, Thanh Cong Bui

**Affiliations:** 1Department of Family and Preventive Medicine, College of Medicine, University of Oklahoma Health Sciences Center, Oklahoma City, OK 73104, USA; kinsey-huff@ouhsc.edu (K.A.H.); darla-kendzor@ouhsc.edu (D.E.K.); 2Department of Nutritional Sciences, College of Education and Human Sciences, Oklahoma State University, Stillwater, OK 74078, USA; ashlea.braun@okstate.edu; 3Infectious Diseases Section, Department of Internal Medicine, College of Medicine, University of Oklahoma Health Sciences Center, Oklahoma City, OK 73104, USA; michelle-salvaggio@ouhsc.edu; 4Oklahoma City-County Health Department, Oklahoma City, OK 73111, USA; patrick_mcgough@occhd.org; 5TSET Health Promotion Research Center, Stephenson Cancer Center, University of Oklahoma Health Sciences Center, 655 Research Parkway, Suite 400, Oklahoma City, OK 73104, USA; summer-frankpearce@ouhsc.edu; 6Department of Biostatistics and Epidemiology, Hudson College of Public Health, University of Oklahoma Health Sciences Center, Oklahoma City, OK 73104, USA

**Keywords:** HPV vaccination, people with HIV, HPV-related cancer prevention

## Abstract

People with HIV (PWH) and their sexual partners have increased risk of human papillomavirus (HPV) infection. Despite recommended HPV vaccination for PWH aged 18–26 years, vaccination rates among PWH remain low. This qualitative study used the Information–Motivation–Behavioral Skills (IMBS) model to identify factors influencing the decisions of PWH around promoting HPV vaccination to their sexual partners. Fourteen PWH with diverse sociodemographic characteristics participated in four focus-group discussions. Data were analyzed using thematic content analysis; codes and themes included IMBS constructs. For the information construct, the need for improved HPV education emerged as the driving factor for HPV vaccine uptake and discussing HPV vaccines with partners. Focal reasons for being unvaccinated included low knowledge of HPV risk, asymptomatic cancer-causing HPV, HPV vaccines, and vaccine eligibility. Salient factors in the motivation construct included the preventive benefits of HPV vaccination to both self and sexual partners. Salient factors in the behavioral skills construct included: accessing vaccine, low self-confidence and skills for promoting vaccination, relationships with sexual partners, partners’ vaccine hesitancy, and stigma. Race/ethnicity impacted HPV vaccination promotion; important determinants included perceptions of HPV-related diseases as “White people’s diseases” among Black people, and discrimination against those with HPV-related diseases among the Hispanic population.

## 1. Introduction

Human papillomavirus (HPV) is the most common sexually transmitted infection (STI) [1]. The Centers for Disease Control and Prevention (CDC) estimates approximately 36,500 new cancers in the United States (US) are caused by HPV annually, including 91% of cervical, 75% of vaginal, 91% of anal, and 70% of oropharyngeal cancers [2]. It is well-established that immunocompromised individuals, including people with HIV (PWH), have an increased risk of these HPV-related diseases compared to their immunocompetent counterparts [3]. Accordingly, the CDC recommends routine HPV vaccination for adolescents and young adults (women aged up to 26 years and men aged up to 21 years) as well as men who have sex with men (MSM) and immunocompromised people (including PWH) through age 26 [4]. Clinical trials demonstrate that HPV vaccines are safe, immunogenic, and efficacious in PWH [5,6,7,8,9]. Furthermore, HPV vaccine is equally effective in HIV-negative individuals who are sexually active but not HPV-infected, and HPV-naïve people [5,10]. The US Food and Drug Administration (FDA) approves Gardasil 9 for use in people through age 45 [11].

Despite increased HPV vaccination among adolescents nationwide and in Oklahoma [12,13], general coverage and awareness of HPV vaccines in eligible adults remains low, specifically in high-risk individuals [14,15]. Nationwide, from 2012 to 2018, the percentage of adults aged 18–26 who received the recommended number of doses increased from 13.8% to 21.5%.8, while among MSM aged 18–26 years, coverage ranged from 33% to 38% between 2017 and 2020 [16,17]. Thus, there is a need for HPV vaccination programs targeted to high-risk adult populations.

Previous HPV vaccination promotion efforts have targeted eligible individuals, parents, and healthcare providers [18,19]. However, no study has examined sexual partners’ influence on HPV vaccination uptake. Systematic reviews have shown that patient-led partner referral approaches, in which index patients with diagnosed STIs notify their partners about the need for testing and treatment, is very effective at managing STIs [20,21,22]. Additionally, partner referral is as effective for casual partners as for regular partners [23]. Thus, partner referral may represent an effective tool for HPV vaccination promotion. Using the Information–Motivation–Behavioral Skills (IMBS) model, [24] this qualitative study aims to identify salient factors that might influence the decision of PWH to promote or not promote the recommended HPV vaccine to their sexual partners.

## 2. Materials and Methods

### 2.1. Theoretical Framework

The IMBS model has been used to promote several health behaviors [25,26,27,28,29] and is the theoretical framework for developing the intervention in our parent study. Consistent with the IMBS model, constructs for this study include information about HPV infection and vaccination, motivations to recommend HPV vaccination to sexual partners, and the behavioral skills to recommend HPV vaccination to partners and receive the vaccine.

### 2.2. Population

We conducted four focus-group discussions (FGDs) with 14 PWH recruited from the Infectious Diseases Institute of the University of Oklahoma Health Sciences Center (OUHSC) through clinic staff or patient referrals. Using purposive sampling we selected diverse PWH, including high-risk subgroups (e.g., MSM or people who used drugs), age (≥18), urban/rural residence, ethnicity, and sexual identity to enhance generalizability and relevance of findings for high-risk and under-represented groups. Patients were excluded if they had experienced a medical adverse event due to HPV vaccination, were unable to speak or read English, or had impaired cognitive function. The study was approved by the OUHSC Institutional Review Board (IRB# 9458), and subjects gave informed consent prior to participation.

### 2.3. Procedures

FGDs were scheduled for 60–90 min and occurred in a private meeting room at OUHSC. At least two research staff members were at each FGD: one facilitator and one note taker. Discussions were voice recorded and transcribed verbatim. Participants identified themselves and others using an assigned unique identification number to maintain privacy. The interviewer(s) asked general, IMBS construct-based, open-ended questions such as “What would motivate you to engage in encouraging sexual partners to obtain HPV vaccination?” or “When and how would you talk to them about HPV vaccination?” Potential barriers to discussing HPV vaccination and obtaining HPV vaccines were also explored. One FGD conducted in Spanish was translated to English by a professional transcription and translation service provider.

### 2.4. Data Analysis

Qualitative data were analyzed using a deductive approach to thematic content analysis with the aid of MAXQDA version 20.4.1, VERBI software, Berlin, Germany. One researcher conducted an initial coding of all transcripts to assign themes, and two other researchers reviewed the codes and quotes. Any discrepancies were discussed to establish agreement in coding. Themes were based on the IMBS constructs; other relevant issues/topics, such as stigma, and race/ethnicity, were also identified inductively as they were found in the data.

## 3. Results

### 3.1. Participant Characteristics

The mean age was 48 years (range: 29–65 years) at the time of study. Five identified themselves as women (35%). Among self-identified men, seven identified themselves as homosexual and two identified as heterosexual. Per self-identification, participants were of various races, including White, Asian American, Native American, and African American; two were Hispanic. All participants were HIV positive. Over half of the participants received some college education.

### 3.2. Information Construct

In the information construct of the IMBS model, the need to improve HPV education emerged as the driving factor for HPV vaccine uptake and for talking to partners about HPV vaccines. Specifically, focal reasons for being unvaccinated included lack of knowledge about: HPV risk, asymptomatic cancer-causing HPV strains, HPV vaccines, and HPV vaccine eligibility for high-risk adults. A 65-year-old White, homosexual man spoke about the lack of knowledge regarding HPV risk factors within the general public: “*A lot of people don’t think of themselves as being at risk and it would be great to help educate them*”.

A 48-year old Hispanic homosexual man discussed difficulties in promoting immunocompromised individuals due to misperceptions regarding risk: “*Other people* [who] *don’t feel that they are at high risk think that things are never going to happen to them. They’re never going to get sick, or they say they’re immune to any kind of illness. So, it’s harder to motivate those people because they think that with having their partner, or not having a partner, or protecting themselves, nothing is ever going to happen to them. So, it is more difficult for me to motivate someone who isn’t high risk like me*”.

A 41-year old White, heterosexual woman stated: “*Some people without the education [are] going to think we already have it anyway, so* [why] *get the vaccine right? Let them know there’s more than one strain and even if they already have one strain you can prevent others which can also cause cancer*”. A 52-year old Native American, homosexual man stated: “*I’ve been HIV positive for many years, and I was diagnosed with HPV because I knew nothing about it*, [until] *my doctor said I had it*”. He continued: “*I wish when I was growing up we had sex education. They told you about how not to get pregnant, but they didn’t tell you about HIV and all that stuff… I’ve never had a sexually transmitted disease ever. I come out, boom, I get the big one* [HIV]”. This participant also discussed his decision to have open communication with his children about sexual activity at a young age: “*I talked to my sons when they were growing up. They were 3 years old and I had prophylactics laying around, I had POZ magazines laying around. They read it and I didn’t force it down their throats. I just gave them the option so that they know what they’re getting into*”.

### 3.3. Motivation Construct

Salient factors in the motivation construct included HPV vaccination-afforded prevention of HPV infection/transmission and cancer for both self and sexual partners. A 48-year-old Hispanic, homosexual man stated: “*Yes*, [I would encourage my sexual partners to have the HPV vaccines] *to protect myself. I’m more at risk…I think everyone thinks in themselves first, and then in others*”. A 52-year old White, Asian American, homosexual man spoke about being non-monogamous: “*I don’t have a significant other right now, but if I did have someone, I would want them protected and then in turn that would protect myself.*” He also stated: “*If you love somebody you want them to get it done because the research says that it can prevent cancer and other things for their own health, right?*” A 41-year old White, heterosexual woman relayed her experience with HIV and a motivation to protect her partner: “*Mostly loving them and caring about them. So they won’t have to go through some of the stuff we had to go through. Especially from a female perspective and having to do procedures that I had to have done due to* [HIV], *and I think it’s important that the person you’re with is vaccinated because they may have a different strain and then that can cause more abnormal cells and more procedures or, you know, potential cancer*”.

### 3.4. Behavioral Skills Construct

Salient factors related to the behavioral skills construct included the unavailability or inaccessibility of vaccine, lack of self-confidence and skills to talk about HPV vaccination with sexual partners, the relationship with sexual partners, and partners’ vaccine hesitancy. A 45-year-old White, heterosexual man reported: “*X.* [a friend] *told me she wasn’t able to get it from her primary doctor yet, because* [it’s] *not available… Only the county health department has it… I guess it’s early stages* [of] *availability*”.

Several participants also listed associated cost of and transportation issues regarding having the HPV vaccination as barriers. A 41-year old White, heterosexual woman stated: “*I think the people that are already on food stamps or Soonercare* [Oklahoma’s state Medicaid agency] *or* [of low socioeconomic status] *or high risk should automatically be able to get it without worrying about cost. Because I can honestly say that I think if there’s a cost associated with it, most people probably won’t do it because of that*”.

Some patients reported low self-efficacy in talking to their sexual partner(s) about HPV vaccination. A 35-year-old African American, heterosexual woman stated: “*I think that this will be better received after an initial talk with a caseworker or case manager with partners available. I myself as a partner have a certain level of understanding of what HPV is*”. She also reported emotional relationships with sexual partners as a barrier: “*I’m married and that makes it easier for us to communicate. But, at that moment, I couldn’t hear you because I was emotional and mad at you. So, I think a lot of times a neutral third party can always get the conversation started*”. However, a 65-year-old White, homosexual man felt bringing a third party would hinder the conversation: “*My concern with bringing extra people into the conversation is that you may also be inhibiting the person that needs to hear the message from their partner directly, they may feel embarrassed. It’s inappropriate to share that information publicly*”.

A 48-year old Hispanic homosexual man stated his preference for promotion to regular partners over casual partners: “*Well, my stable partner would be the one that I have that comfortability with. It depends if I feel comfortable with them. I don’t know how he would react either, but I think I could tell him. As I said, if it’s with casual relationships, that trust would be a barrier*”.

Several participants addressed vaccine hesitancy as a barrier. A 41-year-old White, heterosexual woman stated: “*Some people are opposed to vaccines in any form. Unfortunately, that’s just a barrier that sometimes we just can’t do anything about right now… I have several friends that are like that. They don’t get their kids vaccinated and things like* [that] *that deal with the government conspiracy. Like my cousin’s wife she is a chiropractor and she feels like they’re not necessary. She believes in like holistic medicine and things like that*”.

### 3.5. Stigma

Stigma toward sexuality-related issues central to the topic of HPV vaccines emerged as a barrier to both HPV vaccination discussion and uptake. A 52-year old Native American, homosexual man related the stigma behind their sexual orientation and HIV+ status with HPV: “W*hen I so-called “came out,” half my life was over because I had never acknowledged that I was gay. I got* [HIV] *positive pretty quick and we’ve always had a group called “Oklahoma’s Living Positive,” so there’s a whole range of people and the more you talk about it the better it is… I guess it’s probably the same with HPV*”. A 41-year old White, heterosexual woman discussed the current vaccine eligibility recommendations for high-risk individuals and associated stigma: “[The] *HPV vaccine is recommended through age 26 for all women, men who have sex with men, bisexual, transgender and people with HIV. But what about* [heterosexual] *men through age 26?… First of all, I feel it would cause the stigma. It’s not right. Second… that means that all the promiscuous heterosexual male and females are going to say, ‘I’m not in that group,’…but they’ve already been exposed to* [their] *partner*”.

### 3.6. Race and Ethnicity

Race/ethnicity also emerged as a factor affecting HPV vaccination discussion and uptake. One 35-year-old African American woman spoke about her reason for participating in the study: “*I’m always willing to participate because there’s not many heterosexual African American women who get to share their opinions about their status and how they feel about their status and research on how to get better. I want to be an advocate for myself so I won’t be left out of the study*”. She also explained that many African Americans think that HPV does not affect them: “*It’s been marginalized as this one thing, and African Americans think it’s a rich White person thing. Like ‘oh please girl put a condom on and I’m not going to tell you again.’ So you’ve got to appeal to every group… It touches everybody, it doesn’t matter.* [People think that HPV is a] *rich White people’s disease, and I don’t personally believe like that, but there are people who would be like ‘child please we are Black we don’t need that.’ And we* [are] *the main ones dying from stuff because we believe we don’t get that*”. A 48-year-old Hispanic, homosexual man spoke about the discrimination faced in the Hispanic community: “*And also, if it’s in the Hispanic community, as I said, everything is more difficult. Well, like us—well, for me—thinking about the illness, sometimes it’s not so much the illness but the discrimination against you*”.

## 4. Discussion

This study used the IMBS model to identify salient factors influencing the decision of PWH to promote or not promote the recommended HPV vaccine to their sexual partners. Given low HPV vaccine uptake and awareness among adults, including high-risk populations and PWH, and promising data regarding the role of patient-led partner referral approaches, this is a critical area of inquiry to meet the needs of PWH [14,15,20,21,22].

For the information construct, better education regarding both HPV risk and HPV vaccines were identified as key factors influencing decisions to promote HPV vaccines. These findings are consistent with previous studies examining perceptions of HPV and attitudes towards HPV vaccines in high-risk populations [25,26]. Among MSM, a general “lack of concern” regarding HPV and associated cancer risk has been reported [26]. However, MSM with greater HPV-related knowledge are more likely to be vaccinated, highlighting the importance of information improvement and dissemination [27]. Other populations report beliefs that HPV vaccines are only available for teenagers [28]. Although healthcare workers are identified as the critical catalyst for promoting HPV vaccination, some high-risk individuals report a perceived inability to openly discuss topics related to same-sex relationships [26]. In this study, the importance of improved open communication and education regarding HPV prevention, especially earlier in childhood, was a consistent topic of discussion. Prior studies examining perceptions of HPV vaccines among homosexual, male PWH report current vaccine efforts targeting adults are “too little, too late” [29]. In some instances, the absence of HPV knowledge or awareness may be secondary to previous prioritization of other STIs (notably, HIV) [25]. In 2020, among those aged 13–20 years, only 25% had heard about the HPV vaccination and only 33% were found to have been informed about the association between HPV and cervical cancer specifically [30].

For the motivation construct, preventive benefits of HPV vaccination for both self and sexual partners were key, including preventing the spread and limiting the health consequences of HPV for family and loved ones. This contrasts with recent reports regarding other vaccines in the context of COVID-19, vaccine safety concerns, and increased vaccine hesitancy, which indicate emphasizing individual benefit is more efficacious than prosocial benefit when targeting vaccine hesitancy and uptake [30,31,32]. In contrast, work specifically in adolescent populations and HPV vaccines shows that emphasis on both individual and prosocial benefits contribute to vaccine motivations [33]. Such differences between specific vaccines may relate to the acknowledged risk of transmission of HPV in the general population, as opposed to COVID-19 where multiple misconceptions remain [34]. Multiple population-specific attributes secondary to existing risk factors or demographics may further influence vaccine uptake [35,36,37,38,39,40,41].

For the behavioral skill construct, lack of self-confidence and/or skills to talk about HPV vaccination with sexual partners, relationships with sexual partners, and partners’ vaccine hesitancy were key factors. In addition, issues related to procurement of the vaccine, including perceived unavailability or inaccessibility, were barriers to vaccine uptake; barriers specifically related to transportation and cost were also identified, consistent with prior studies, particularly among individuals of low socioeconomic status [35,42,43,44,45]. Cost was specifically identified in this study as an especially detrimental barrier for high-risk populations.

Lack of confidence and/or skills to talk about HPV vaccination with sexual partners results in reliance on third parties to promote vaccine uptake to loved ones. Sexual minority men reportedly do not typically discuss HPV vaccination with sexual partners, and may be responsive to and reliant upon healthcare providers, who may or may not bring up vaccination [26,27,46,47]. Unfortunately, healthcare providers, and certain healthcare settings (e.g., lesbian, gay, bisexual, transgender, or queer [LGBTQ] friendly settings), can be either facilitators or barriers to vaccine discussions [47,48]. Although healthcare communication regarding vaccination is critical to promotion efforts, this approach has been weakened by COVID-19, and thus communication with partners is a critical reinforcement strategy [49]. Our findings indicate a preference for promotion to stable partners over casual partners due to lack of trust. Previous data indicate MSM who have unprotected anal sex or multiple sex partners tend to have low acceptance of the HPV vaccine, thus, those at highest risk may be less likely to bring up vaccines with their partners [34]. Collectively, this raises concerns regarding limited exposure of those at highest risk to information about vaccination. Furthermore, individuals may be vaccine hesitant and approaches to target this hesitancy among patients or peers are ill-defined [35]. However, the use of “dialogue-based” interventions shows promise for addressing vaccine hesitancy and anti-vaccine rhetoric, both in general and specifically HPV vaccines [50,51,52].

Stigma and race/ethnicity were also identified as barriers to HPV uptake and promotion independent of the IMBS model. Stigma regarding sexual orientation and HIV/HPV status were barriers to promotion. This was particularly concerning for specific groups recommended by the CDC to receive HPV vaccination (i.e., MSM or PWH) and the potential to reveal a vaccine recipient’s status. Such stigma limits willingness to discuss sexuality, or related health concerns, with healthcare providers, in part based on culturally insensitive healthcare [26,53,54,55]. Furthermore, fear of stigma is greatest in those unaware of HPV vaccine availability for men and/or adults, highlighting potential interactions between stigma, race/ethnicity, and IMBS constructs (e.g., information) [28,56]. The misconception identified in this present study that HPV is a “rich White person’s” disease is a possible barrier for vaccination in the African American community in particular. Previous data show African American populations are open to vaccination, particularly for cancer prevention, but voice poor awareness and experiencing stigma, while vaccine uptake is poor across minority populations generally [15,57]. Discrimination regarding HPV status in the Hispanic population was also apparent; healthcare providers are less likely to discuss HPV vaccines with Hispanic populations whose preferred language is Spanish [58], despite healthcare provider recommendation being as critical for uptake in this population as others [59]. Hispanic MSM voice openness to discussing HPV vaccines with LGBTQ-friendly healthcare providers, though not without concerns of feeling overwhelmed if coupled with conversations about HIV, as well as fearing stigma [60]. Thus, inclusive initiatives and addressing medical mistrust are important for improved outcomes in these communities.

This study has numerous strengths, including our purposive sampling strategy to specifically target high-risk and potentially under-represented groups. It also fills a critical gap in the literature considering the recently expanded recommendations with respect to HPV vaccination (e.g., approval of Gardasil 9 for use in men and women through age 45). However, it is not without limitations. These include, most notably, a small sample size, as well as a final sample which was restricted to adults in Oklahoma. These factors may limit generalizability to other populations, and future work should expand upon these data.

Collectively, these data indicate that among PWH, there remains a need to improve awareness of HPV vaccines, while helping overcome both financial and logistical barriers. Our data corroborate previous work highlighting the importance of healthcare providers or other third parties in facilitating HPV vaccine uptake, by overcoming stigma and/or vaccine hesitancy. However, presently, healthcare provider interactions remain insufficient to address gaps in vaccination given associated stigma, and lack of awareness among healthcare providers and non-inclusive healthcare settings. Previous evidence supporting partner referral-based interventions in PWH highlights a potential avenue for improved vaccination in this high-risk population [61,62]. Since previous data indicate informational interventions are insufficient to lead to sustained public health impacts on vaccination, combined approaches leveraging partner referrals, improved healthcare provider interactions, and improved access should be considered to be future interventions [63].

## 5. Conclusions

To promote HPV vaccination among PWH, it is important to address barriers, improve HPV-related education, overcome HPV-related misconceptions, and equip PWH with communication skills to discuss HPV vaccination with both their sexual partners and other members of the PWH community. This should be coupled with other approaches (e.g., healthcare provider communication) for optimal impact.

## Data Availability

The data presented in this study are available upon request. The data are not publicly available due to ethical consideration and the institutional requirement to establish a data sharing agreement before the data can be shared.

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
