# Peer review of "Promoting HPV Vaccination in People with HIV: Factors to Consider"

_ijerph, 2023, doi:10.3390/ijerph20075345_

Round 1

Reviewer 1 Report

The present manuscript reports about a quantitative study conducted through IMBS (Information-Motivation-Behavioral skills) model for identifying the decisive factors influencing the decisions of people with HIV with respect to promoting HPV vaccination to their sexual partners. 

Following are the comments to highlight certain limitations after review of the paper;

1.     The English language needs improvement with respect to grammar and sentence formation.

2.     The participant size for proper data analysis in the creation of model is quite less.

3.     Kindly explain what is the reason behind selecting purposive sampling? The data could have been made more robust and representative of the population if more participants would have been selected in general PWH population and different risk groups.

4.     The population does not represent complete age group and specifically not in the age group of 18-26 years which comes under the recommendation of HPV vaccine and sexually active.

5.     Kindly define population in detail, preferably in a table clearly representing different factors included in the sampling criteria.

6.     The female population involved in the study is quite less.

7. Line 153 - One of the factors incorporated in Behavioral skill construct is unavailability or inaccessibility of vaccine? Is it relevant to incorporate it in behavioral construct instead of motivation construct or information construct?

Author Response

  1. The English language needs improvement with respect to grammar and sentence formation.

- We appreciate this input, and have given the full manuscript a thorough review and made edits accordingly to improve grammar, sentence structure, and overall readability of the manuscript text. 

  1. The participant size for proper data analysis in the creation of model is quite less.

- Although the number of participants is low, we feel the goal of this qualitative study was to identify patterns in participant answers with in-depth focus group discussions. We feel we were able to gain sufficient results and identify common themes. Future studies could implement the salient factors discussed here in the creation of larger, more generalizable research. 

  1. Kindly explain what is the reason behind selecting purposive sampling? The data could have been made more robust and representative of the population if more participants would have been selected in general PWH population and different risk groups.

- We have chosen purposive sampling in order to perform in-depth interviews with a particular emphasis on those known to be high risk (e.g., men who have sex with men (MSM)) as well as those who may be underrepresented in current literature (e.g., rural versus urban residents). This approach was chosen in order to ensure generalizability of findings, while providing data representative of those who may be understudied or worthy of particular emphasis in future HPV prevention efforts, even if this means a resulting sample size that is lower, but less representative.  We have added clarifying language in lines 88-89.

  1. The population does not represent complete age group and specifically not in the age group of 18-26 years which comes under the recommendation of HPV vaccine and sexually active.

- We appreciate this observation, and it points to an important need for this study given the newest adult HPV vaccination recommendations from the FDA/CDC recommendations are for 18-26 and 27-45 years of age in those at high-risk of HPV infection or who might otherwise benefit. Accordingly, this study includes PWH over the ages 18-26 to include these high-risk populations (people living with HIV/AIDS). We have made changes in lines 51-58 to clarify this justification. In addition, some individuals included in this study are above the age recommendation (e.g., 65 years) however we included them as they provide useful insight regarding the target populations and given their familiarity with the issue/population at hand.

  1. Kindly define population in detail, preferably in a table clearly representing different factors included in the sampling criteria.

- We will gladly provide further information regarding our inclusion/exclusion criteria and have added this to lines 89-91. We used purposive sampling to select a sample that represents different targeted populations (e.g., PLWH, MSM, bisexuals) and that is diverse with regard to age, residence, ethnicity, and sexual identity. These participants were recruited from the OUHSC Infectious Diseases Institute (IDI) per recommendations of the clinic supervisor or through referral. To be included in the FGDs, potential participants must have been aged ≥18 years and able to communicate in English. Patients were excluded if they had experienced a medical adverse event due to HPV vaccination, were unable to speak or read English, or had impaired cognitive function. Lines 130-135 include patient characteristics in our study.

  1. The female population involved in the study is quite less.

- Per our purposive sampling strategy (as outlined in lines 86-89), we specifically targeted MSM as one high-risk subgroup. Thus, this particular group is appropriately oversampled and the resulting sample includes more males than females.

  1. Line 153 - One of the factors incorporated in Behavioral skill construct is unavailability or inaccessibility of vaccine? Is it relevant to incorporate it in behavioral construct instead of motivation construct or information construct?

- We appreciate this attention to detail and thoughtfulness. As the behavioral construct within the IMBS model relates to skills necessary to execute a behavior, we conceptualize these conveyed issues (e.g., availability and accessibility) as those central to the ability to execute procurement skills. Further, if we think of the potential strategies designed based on the IMBS model, strategies focused on education/information or motivation are not as relevant for these issues, whereas those focused on enhancing skills (i.e., helping individuals procure the vaccine if it is perceived that availability is low) are more appropriate, pointing to the consistency with the behavioral construct. We have made some minor modification to the text (lines 305-306) to clarify this perspective.

Reviewer 2 Report

The authors presented the answers from HIV patients about HPV vaccination.  The results seem just list patients' answers and are highly subjective. I wonder which result using the analysis described in Material and method. The manuscript should be more scientific and clear. It is hard to understand the authors' points immediately since the data was not well organized.  

Author Response

The authors presented the answers from HIV patients about HPV vaccination.  The results seem just list patients' answers and are highly subjective. I wonder which result using the analysis described in Material and method. The manuscript should be more scientific and clear. It is hard to understand the authors' points immediately since the data was not well organized. 

- We appreciate this perspective and have added some clarifications throughout the results. However, we want to acknowledge that all of the results are those obtained utilizing the qualitative analysis method described in lines 122-127 (i.e., deductive thematic content analysis an inductive analysis as appropriate with a total of three coders), which were selected to remove unnecessary subjectivity and bias.

Reviewer 3 Report

Promoting HPV vaccination in people with HIV: Factors to consider

Comments

Thank you for the opportunity to review this qualitative study. This research used Information-Motivation-Behavioral Skills (IMBS) Model to identify the factors that influence HPV vaccination recommendation to the sexual partners in the HIV positive people.

This research will help to drive the research on this topic forward. However, through some observations and suggestions I would like to suggest a major revision to make the manuscript stronger

  Major Point

1.    I would have liked to have seen the questionnaire and supporting result files attached as supplemental files.

Title:

1.    Adequate

Abstract:

Adequate

Introduction

1.    Adequate

Materials and Methods

1.    Questionnaire and supporting result file in the supplemental section are missing.

Results

1.    Adequate

Discussion

1.    Since it is a narrative study with a small sample size so it lacks generalizability to accurately address the outcome of HPV vaccination hence the author should elaborate the limitation and future prospects related to the study.

 Conclusion

1.    Adequate

References

1.    Correct the details (volume, issue, page) of Ref# ,28,41,43,46,63

Author Response

  1. I would have liked to have seen the questionnaire and supporting result files attached as supplemental files.

-We appreciate this request and are understanding of transparency for inclusion of a questionnaire. We utilized open-ended questions during the focus group discussions and did not provide a specific questionnaire to our participants for our data collection. All focus group discussions included questions such as “What would motivate you to engage in encouraging sexual partners to obtain HPV vaccination?” and “What would motivate you to encourage your main sexual partners, like a spouse or a long-term partner, to obtain HPV vaccination?” or “Is there anything different that would motivate you to encourage casual sexual partners to obtain the HPV vaccine?” Barriers to HPV promotion were addressed    throughout the discussions.

  1. Since it is a narrative study with a small sample size so it lacks generalizability to accurately address the outcome of HPV vaccination hence the author should elaborate the limitation and future prospects related to the study.

- Thank you for this thoughtful comment. Although the number of participants is low, the purposive sampling strategy employed to specifically target high-risk and underrepresented groups enhances generalizability in order to capture the unique experiences and perspectives of these groups.  This does, however, point to an important limitation that we have addressed in lines 357-364

  1. Correct the details (volume, issue, page) of Ref# ,28,41,43,46,63

- We appreciate your attention to detail and have made these corrections to the references above.

Round 2

Reviewer 1 Report

Thank you for clarifying the raised concerns in the revised manuscript.

Reviewer 2 Report

The manuscript has improved. While the data is still limited, The presented data supported the author's opinion. Thank you for the answers to reviewers.